

# Cluster-based ensemble means for climate model intercomparison

Richard Hyde[1], Ryan Hossaini[1], and Amber Leeson[2,1]

[1]Lancaster Environment Centre, Lancaster University, Lancaster, LA1 4WA, UK
[2]Data Science Institute, Lancaster University, Lancaster, LA1 4WA, UK

*Correspondence to:* Richard Hyde (r.hyde1@lancaster.ac.uk), Ryan Hossaini (r.hossaini@lancaster.ac.uk)

**Abstract.** Clustering – the automated grouping of similar data – can provide powerful and unique insight into large and complex data sets, in a fast and computationally-efficient manner. While clustering has been used in a variety of fields (from medical image processing to economics), its application within atmospheric science has been fairly limited to date, and the potential benefits of the application of advanced clustering techniques to climate data (both model output and observations) may yet to be fully realised. In this paper, we explore the specific application of clustering to the calculation of multi-model means from climate model output. A standard rudimentary approach to multi-model mean (MMM) calculation simply involves taking the arithmetic mean of all models in a given ensemble, over a particular space/time domain (a one model one vote approach). We hypothesise that clustering can provide a useful data-driven method of (a) excluding 'poor' model data from MMM calculations, on a grid-cell basis, thus (b) maximising retention of 'good' data, and avoiding the blanket exclusion of models, where appropriate. We focus our analysis on chemistry-climate model (CCM) output of tropospheric ozone – an important greenhouse gas – from the recent Atmospheric Chemistry and Climate Model Intercomparison Project (ACCMIP). Cluster-based MMM fields of tropospheric column ozone were generated from the ACCMIP ensemble using the Data Density based Clustering (DDC) algorithm. The cluster-based MMM was compared to the simple arithmetic MMM (one model one vote approach) and each MMM was evaluated against an observed satellite-based tropospheric ozone climatology, as used in the original ACCMIP study. As a proof of concept, we show the proposed clustering technique can offer improvement in terms of reducing the absolute bias between the MMMs and observations. For example, the global mean absolute bias from the cluster-based MMM is reduced in all months, up to ∼15%, compared to the simple arithmetic MMM. On a grid-cell basis, the bias is reduced at more than 60% of all locations. Some locations are found to be unaffected by the clustering process, while in others the bias increases, albeit slightly. This and other caveats of the clustering techniques are discussed. Finally, while we have focused on tropospheric ozone, the principles underlying the cluster-based MMMs are applicable to other prognostic variables from climate models. We further demonstrate that clustering can provide a viable and useful framework in which to assess and visualise model spread, offering insight into geographical areas of agreement between models and a qualitative measure of diversity across an ensemble.



## 1 Introduction

Clustering is a flexible and unsupervised numerical technique that involves the segregation of data into statistically similar groups (or "clusters"). These groups can either be determined entirely by the properties of the data itself, or guided by user constraints. Numerous clustering algorithms have been developed, each with varying degrees of complexity. The k-means
clustering algorithm, for example, is a relatively simple and popular technique used in several atmospheric science problems (e.g., Mace et al., 2011; Qin et al., 2012; Austin et al., 2013; Arroyo et al., 2017). Specifically related to climate science, clustering has also been used for automated classification of various remote sensing data (e.g., Viovy, 2000), the interpretation of ocean-climate indices and climate patterns (Zscheischler et al., 2012; Yuan and Wood, 2012; Bador et al., 2015), in describing spatiotemporal patterns of rainfall (Muñoz Díaz and Rodrigo, 2004), and to classify surface ozone measurements from a large
network of sites (Lyapina et al., 2016), among several other applications.

An area where the applicability of clustering has yet to be fully explored is in the analysis of model output from multi-model intercomparison initiatives, involving chemical transport models (CTMs), climate models, or chemistry-climate models (CCMs). Such initiatives are now common and form an integral part of scientific assessment of atmospheric composition, particularly in international policy-facing research concerning climate change. For example, recent model intercomparison
studies have considered stratospheric ozone layer recovery (Eyring et al., 2010), the climate impacts of long-term tropospheric ozone trends (Young et al., 2013; Stevenson et al., 2013), and paleoclimatology (Braconnot et al., 2012), among others.

In virtually all multi-model intercomparisons related to atmospheric composition, the multi-model mean (MMM) of a given prognostic variable is computed for a given space/time domain, commonly reported along with the model spread, or MMM standard deviation ($\sigma$). In most instances, the MMM is computed from a simple arithmetic mean of all models (i.e. a one
model one vote approach), such as during the recent Atmospheric Chemistry and Climate Model Intercomparison Project (ACCMIP) studies of tropospheric ozone and the hydroxyl radical, OH (Young et al., 2013; Voulgarakis et al., 2013). For chemical species such as these, that exhibit large space/time inhomogeneity in their tropospheric abundance, rarely a single model will be universally best performing (i.e. at all locations/times). In this regard, a MMM is a useful quantity and is often considered a *best estimate* that includes robust features (that are still apparent after averaging) from the ensemble of models.
More involved MMM approaches may somehow weight individual model contributions (e.g., DelSole et al., 2013; Haughton et al., 2015; Wanders and Wood, 2016), for example based on their performance against a set of observations, thus potentially diluting spurious features from individual models – though such approaches have been somewhat rarely implemented in recent CCM intercomparisons. Furthermore, it is not uncommon for individual models to be excluded entirely from the MMM if deemed particularly poor on the basis of an evaluation against a set of observations (e.g., Hossaini et al., 2016), or if deemed a
clear/substantial outlier with respect to the majority of other models (e.g., Eyring et al., 2010).

In this study, we hypothesise that clustering techniques can provide a flexible data-driven method of including/excluding selective model data into MMM calculations, providing (a) a data-driven method for retention of "good" model data, relative to blanket model exclusions, and (b) potential insight into model development needs. We focus our analysis on tropospheric column ozone data from 14 atmospheric models (mostly CCMs) that took part in the ACCMIP intercomparison (Young et al.,





2013). Our specific objectives are to (i.) derive a cluster-based MMM field of tropospheric column ozone, (ii.) evaluate the performance of the cluster-based MMM against more rudimentary MMM approaches by comparison to observations, and (iii.) explore the use of clustering as a tool to identify and visualise diversity across a model ensemble, with the potential to inform model development. We demonstrate that clustering can reduce the overall bias between modelled (i.e. MMM) and observed

tropospheric column ozone, while maximising data retention from individual models. Advantages of the clustering approach over more traditional weighting methods are discussed, as are limitations of the techniques and areas of future development.

The paper is structured as follows. Section 2 provides a brief overview of cluster-based classification. Section 3 describes the principles of the proposed clustering technique, exemplified using an idealised synthetic data set. Section 4 describes the application of the clustering techniques to multi-model output from the ACCMIP intercomparison. Results from the ACCMIP

clustering and discussion are presented in Sect. 5. Recommendations for future research are given in Sect. 6 and our conclusions are given in Sect. 7.

## 2   A brief overview of cluster-based classification

Clustering is a well established technique for the unsupervised grouping (classification) of similar data. The unsupervised nature of clustering overcomes many of the traditional short-comings of classification techniques, e.g. no a-priori information

is required, classes (clusters) are data-driven and may adapt to underlying changes in the data relationships. Many offline clustering algorithms are available and no single algorithm can be considered the 'best' for all situations. Several in-depth reviews of clustering techniques have recently been published (Aggarwal and Reddy, 2014; Nisha and Kaur, 2015; Xu and Tian, 2015), therefore here we outline only briefly the features of some common techniques, in the context of this work.

Perhaps the most popular method employed within atmospheric science is the k-means clustering algorithm (MacQueen,

1967). K-means generates hyper-elliptical, unconstrained, clusters offering the benefit of fast processing and a constrained number of clusters. However, the method requires that the number of clusters is specified beforehand, limiting its usefulness in data mining and often means that the techniques results in clusters that fit the "required answer". Other algorithms that do not require prior knowledge of the data clusters, and are therefore considered to be more data-driven, include subtractive clustering. This generates the required number of clusters, though is limited by a maximum cluster radius, thereby potentially dividing

natural groups of data. This technique can also be prohibitively slow where large data sets are involved, as calculations are repeated for all remaining data samples after each cluster is formed.

Recently, purely data-driven techniques have been developed, including grid-based algorithms and density-based algorithms. Many of these recent developments can match, or exceed, the older techniques for speed, consistency, "accuracy" and have the added ability to be data-driven with minimal user intervention. As such, these techniques have the potential to provide powerful

semi-automated insight into large data sets, such as output generated from atmospheric models, or a large ensemble of models. In this study, we use the Data Density based Clustering (DDC) algorithm (Hyde and Angelov, 2014). The underlying principle is that data classified into a DDC-generated cluster is more similar to other data within said cluster, than it is to data within other clusters. The DDC algorithm has the advantage in that the scope of each cluster is well defined. For example, maximum





distances can be set, in the physical world as well as in the data space, which define the spatial regions covered by clusters and the range of data values to be considered similar. DDC matches simple techniques such as k-means for speed, but requires no prior information on the number of clusters. It is also robust to using larger cluster radii, as the algorithm adjusts the radii to match the data contained within the cluster. A simple application of the algorithm is described in Sect. 3 below.

## 3   The principles of cluster-based multi model means

In this section we explain the principles behind the proposed technique for generating cluster-based MMMs, using a simple synthetic data set as an example. Application of the technique to real data is more complex and is detailed in later sections.

Chemistry-climate models attempt to simulate the atmospheric distribution of numerous chemical compounds including, for example, tropospheric ozone. Model skill/performance is typically assessed by comparison to atmospheric observations made at discrete times and locations. For a given comparison, a model may exhibit a phase offset in time or space, resulting in a large model-measurement bias, suggesting an inaccurate model - perhaps due to a process-level deficiency. However, in some cases phase offsets in space, for example, could be related to a sampling or 'mismatch' error, particularly when comparing output from coarse resolution models to point source observational data. Such errors are commonly encountered in inverse modelling studies, for example, that aim to derive top-down emissions of a given compound based on atmospheric observations (e.g., Chen and Prinn, 2006). To account for such, a flexible technique that looks beyond a specific space/time and that can identify similar data in the surrounding data space is required. To illustrate this we use a simple 2D synthetic data set as shown in Figure 1.

The data shown in Figure 1 includes synthetic 'observations' (panel a) generated using a sin function. The values on the $x$ and $y$ axes are arbitrary and the data is intended to mimic a generic observation that is spatially non-uniform. We also consider 4 different sets of synthetic 'model' data (panel b) which, with respect to the observations, exhibit (1.) a small consistent positive bias (red), (2.) a small consistent negative bias (dark blue), (3.) a large bias (green), and (4.) a slight phase offset (cyan); clearly model 3 would be considered a poor/outlier model. Taking the 4 models to be an ensemble, a simple MMM is generated by taking the arithmetic mean of the 4 model data sets at each location (i.e. no clustering involved). We also apply the DDC algorithm to the data, as shown in panel (c), to generate a cluster-based MMM. The ellipses represent the different clusters that are formed which, as noted, can extend to nearby surrounding data space.

The DDC-based MMM is calculated by taking the mean of the data in the dominant cluster at each location. A cluster is considered dominant if it contains the most data samples. We therefore assume that this is the most likely region to contain the observed value. For example, with reference to panel (c), it can be seen that a cluster is formed at ∼x=0.4, ∼y=-0.8. Data within this cluster is not included in the MMM at this location, as a more populous cluster at the same location (∼0.4, ∼0.6) is present.

Panel (d) of Figure 1 compares each MMM to the observed data; the simple arithmetic MMM (one model one vote approach) provides poorer agreement compared to the cluster-based MMM, largely due to model 3 being included in the mean calculation





for the former. Note, each MMM is independent of the observations and in this regard the process is analogous to a multi-model prediction of a future variable (i.e. with no observational constraint).

# 4 Application of clustering to ACCMIP model data

## 4.1 Overview of ACCMIP datasets

The Atmospheric Chemistry and Climate Model Intercomparison Project (ACCMIP) was a multi-model initiative conducted to investigate the atmospheric abundance of key climate forcing agents, including tropospheric ozone, and their change over time (e.g., Young et al., 2013; Stevenson et al., 2013; Lamarque et al., 2013). For our purposes, we use the ACCMIP climate model data as an example of a typical multi-model ensemble on which to perform the clustering. A benefit of using ACCMIP output is that the data has been extensively handled and analysed by various groups, allowing direct comparison of our findings with

published work, and the data is publicly available. We focus our analysis on modelled tropospheric column ozone data (Dobson Units) generated by 14 of the ACCMIP models (see Table A1). A detailed description of the models and their underlying processes can be found in the above ACCMIP publications. For each model, we analyse output from the *historical* simulation corresponding to the year 2000 (Young et al., 2013). Within ACCMIP, evaluation of models and the MMM was performed by comparison to a tropospheric ozone column climatology based on Ozone Monitoring Instrument (OMI) and Microwave Limb

Sounder (MLS) satellite measurements (Ziemke et al., 2011). The monthly climatology extends from 60°N to 60°S. Following Young et al. (2013), we compare MMMs (generated either with clustering or without) to the observed climatology within this latitude range.

## 4.2 Cluster based ensemble means

Here we illustrate the procedure by which cluster-based MMMs are generated using the DDC algorithm. The cluster-based

methods are independent of observations but require a *predicted truth*. The predicted truth is a fundamental concept as it influences the decision of the algorithm on whether to include/exclude data from a given model, at a given location, into the MMM. The predicted truth can be generated in one of two ways. The first is a simple arithmetic mean of all the model data at a given location. The second is the average of the model data within $1\sigma$ of the arithmetic mean, referred to as the "sigma-mean" predicted truth. Note, clearly both of these techniques can be used without clustering, with the former retaining 100% of the

data in a MMM and the latter being essentially a data reduction technique. The schematic given in Figure 2 illustrates how the predicted truth is used by the DDC clustering algorithm. Again, arbitrary synthetic data is first used to exemplify the key principles. The synthetic data represents output from 8 different models that together form an ensemble (i.e. the 8 different connected lines). The ellipses represent different clusters that are assigned A-E. Again, we note the flexibility of the clusters in looking at the surrounding data space for similar data.

For DDC generated clusters, at a given location the MMM of the ensemble is calculated as an average of all data in the cluster that contains the predicted truth (red diamonds). If no suitable cluster is nominated, then the predicted truth value for





that location is used. For the example data shown in Figure 2, the data used for the DDC MMM at locations 1, 2 and 3 will be the arithmetic mean of the data from clusters C, C and C. This is because at each of these locations, the predicted truth lies in cluster C. Thus, the models included at locations 1, 2 and 3, are those denoted by the the following colours: (green, cyan), (green, cyan), (green, cyan, purple, yellow). In this way, data farthest from the mean (predicted truth), i.e. not in agreement
with other models, is removed locally.

### 4.3  Initialisation of clustering algorithms

Initialisation of the clustering algorithms involves selecting suitable initial cluster radii for each of the data dimensions, in this case: longitude, latitude and column ozone. In this work, we operate the clustering on a spatial basis only, in order to account for spatial mismatches as discussed in Sect. 3. When selecting these radii, it should be noted that the clustering algorithms
perform best with data on a similar scale in each axis. To this end we scale the data to approximately 0-1 in each dimension.

#### 4.3.1  Ozone radius selection

Modelled ozone values are scaled to approximately 0-1 using the average minimum value and average range of the data in each month as given by Eq. (1):

$$O_{3S(m,i,t)} = \frac{12 O_{3(m,i,t)} - \sum\limits_{t=1}^{12} min(O_{3(*,*,t)})}{\sum\limits_{t=1}^{12} max(O_{3(*,*,t)}) - \sum\limits_{t=1}^{12} min(O_{3(*,*,t)})} \qquad (1)$$

Where $O_3$ and $O_{3S}$ are the modelled and scaled ozone values, respectively, at location, '$i$', as estimated by model, '$m$', at time '$t$'.

The initial ozone cluster radius is taken to be the average of twice the standard deviation on the model spread, Eq. (2):

$$r_{O_3} = \frac{2 \sum\limits_{i=1}^{n} \sum\limits_{t=1}^{12} SD(O_{3(*,i,t)})}{12n} \qquad (2)$$

where $SD(O_{3(*,i,t)})$ is the standard deviation of the ozone values of the ensemble at time '$t$' at location '$i$' and '$n$' is the
number of grid spaces.

This corresponds to an initial radius of 8.3 DU (0.1523 when scaled as in equation 1). Note, the cluster radii evolve in a data driven manner, excluding outliers and extreme values from the clusters. In consequence, final cluster radii using DDC range from 0.1-8.3 DU, with 70% of the clusters actually used in model selection for the MMM calculation having a radius <7 DU (Figure A1). This radius is indicative of the range of $O_3$ data at each grid location, after outliers have been removed by the
clustering process.





### 4.3.2 Spatial radii selection

In later sections we show that our cluster-based MMM column ozone field exhibits a lower global mean absolute bias with respect to observations, compared to the simple arithmetic MMM. This improvement offered by clustering exhibits some sensitivity to the choice of initial radii in the spatial dimensions. In the latitude dimension, the improvement exhibits a negative

correlation with radius (r = -0.88); i.e. improvement lessens with larger radii. Results are presented from here on for an initial cluster radii of 1.5 grid-cells (0.0683 when normalized to 0-1) and 2.5 grid-cells (0.0352) in the latitude and longitude direction respectively, as this combination was found to give the best improvement overall. As in Sect. 4.2.1., the cluster radii evolve in a data-driven manner and final cluster radii range from 1 - 1.6 grid-cells (0.0455 - 0.0728) in the latitude direction, and 1 - 2.6 grid-cells (0.0141 - 0.0367) in the longitude direction. 92% and 99% of clusters used in model selection for the DDC

MMM have a radius of less than or equal to 1.1 grid-cells in the latitude and longitude directions, respectively. A radius of 1.1 grid-cells means that at each location, the cluster used in model selection potentially contains data from that cell and from cells with which it shares a border. While data from nearby grids may affect the location of a cluster, this data is not included in the calculation of the MMM. Rather, the MMM at each location is the mean of the data at that location only, that is included in the nominated cluster.

### 4.4 Scenarios and metrics

Using the principles described above, the DDC algorithm was applied to the ACCMIP model ensemble of tropospheric column ozone, on a monthly basis. As previously noted, the clustering algorithms require a predicted truth that can be calculated on a simple mean or a sigma-mean basis, thus 2 different permutations are possible for our cluster-based MMM. We also calculated MMMs of the same data using a simple arithmetic mean and a sigma-mean, without any clustering involved. In

the subsequent Results sections, we compare each of these MMMs and evaluate their performance by comparison to the satellite-based tropospheric ozone climatology described in Sect. 4.1. In particular, we focus the analysis on whether or not the cluster-based MMMs provide 'improvement' over the most rudimentary approach, the simple arithmetic mean, that omits no model data. In summary, 4 MMMs are considered: (1) Simple MMM, (2) Sigma MMM, (3) Cluster-based MMM (predicted truth = simple mean) and (4) Cluster-based MMM (predicted truth = sigma-mean).

Several metrics are used in the ensuing discussion, including the model-observation mean bias (equation 3), and the absolute mean bias (equation 4), where M and O are the MMM and observed ozone field at location *i*, respectively.

$$\text{Mean Bias} = \frac{1}{n}\sum_{i=1}^{n}(M_i - O_i) \tag{3}$$

$$\text{Mean Absolute Bias} = \frac{1}{n}\sum_{i=1}^{n} \mid M_i - O_i \mid \tag{4}$$



## 5 Results and discussion

### 5.1 Assessment of cluster-based MMM on a global basis

We first evaluate the relative performance of the cluster-based MMM with respect to the simple MMM on a global monthly mean basis. The observed column ozone data (DU) is presented in Table 1, along with equivalent MMM estimates, rows 2 and

5 3, obtained using a simple arithmetic mean approach – as in Table 3 of Young et al. (2013) – and a sigma mean approach. These are followed by cluster-based MMMs obtained from the 2 different scenarios outlined in Sect. 4.4. For each MMM, the mean bias (equation 3) is given in Table 2. Note, the focus of this work is not to evaluate the skill of individual ACCMIP models, or the ensemble as a whole, with regard to underlying chemical processes. For that, an in-depth discussion should be sought from Young et al. (2013). Rather, our focus is to assess the fidelity of the cluster-based MMMs relative to MMMs based on

simpler approaches. Based on Tables 1 and 2 it is clear that the ACCMIP ensemble provide a reasonably good simulation of tropospheric column ozone with respect to the observations, in a global mean sense. For example, the annual mean bias for each of the various MMMs is <1 DU. The cluster-based MMMs exhibit a bias (-0.6 DU) that is marginally greater to that obtained from the simple arithmetic MMM (-0.4 DU). However, note the global mean biases reflect an amalgamation of positive and negative biases, masking important regional/hemispheric differences as outlined below.

Table 3 is similar to Table 2 but presents the *absolute* biases, again on a global mean basis. The cluster-based MMMs exhibit lower biases in all months relative to those obtained from the simple arithmetic mean approach. The 'improvement' offered by the cluster-based MMMs relative to the simple arithmetic MMM is shown in Figure 3; that is, the percentage reduction in the global mean absolute bias. Both cluster-based MMM variants lead to improvements, reducing the MMM global bias by ∼3-16%, depending on the month. While we do not over interpret our findings from a model process standpoint, a distinct

monthly variability is apparent in the bias reduction, with the lowest overall improvement in the months June-August. This is also the case for the (non-clustered) sigma-mean MMM, also shown in Figure 3, which exhibits a negative bias reduction (i.e. actually performs 'worse') with respect to the simple MMM during these months, despite offering a slight improvement overall. From Tables 1 and 2, both the observed annual mean ozone column and the absolute (model-observation) biases are lowest in these months. Based on the latter, it is perhaps unsurprising, therefore, that the improvement offered by clustering in

these months is relatively modest. Recall, the clustering techniques exclude selective data from the MMM at a given location, from a given model, if there is poor agreement with other models in the ensemble. Thus, if all models agree well, regardless of whether their values are accurate or not, few (or no) model data may be removed. In this case, the cluster-based MMMs will not vary substantially from the simple arithmetic MMM and relatively little (or no) 'improvement' (i.e. bias reduction) will be achieved through clustering. A similar situation also arises if the models have a wide spread of values at a given location; data

ignored by the cluster-based MMM may be equally divided above and below the predicted truth (i.e. simple or sigma). In such a case, removing these data will have little effect and the cluster-based MMM will vary little from the simple MMM.



## 5.2 Assessment of cluster-based MMM: spatial variability

We extend the above discussion to evaluating spatial variability in the performance of the various MMMs. Spatial variability of the monthly mean bias (model - observations, DU) for the simple MMM case is shown in Figure 4. A similar figure but for the cluster-based MMM is shown in Figure 5. As was shown in Young et al. (2013), the ACCMIP ensemble tends to exhibit a

5 high bias with respect to the observations in the Northern Hemisphere (NH), and a low bias in the Southern Hemisphere (SH) (Figure 4). The positive and negative biases largely cancel yielding an overall small negative bias when expressed as a global mean (see Table 2). Based on Figures 4 and 5, differences between the simple rudimentary MMM and the cluster-based MMM are difficult to fully discern by eye. The differences are more apparent when viewed as absolute biases, as given in Figures 6 and 7. However, most striking is Figure 8, that compares the improvement, i.e. the reduction in model-observation absolute bias

for the cluster-based MMM, relative to the simple arithmetic MMM. Geographically, clustering provides some improvement at all latitudes, though particularly in the NH and including over central Asia, Europe and the USA - where ozone precursor emissions are generally elevated due to anthropogenic processes. Note, the ACCMIP ensemble overestimates the ozone column climatology in the NH (e.g. see Figures 4 and 6), see also Young et al. (2013), thus effectively the improvement seen in the cluster-based MMM reflects some removal of data at the upper end of the model range (i.e those models with relatively high

ozone). Typical bias reduction is of the order of 1-5 DU, though larger reductions of >5 DU are achieved in both hemispheres in some grid-boxes.

Also apparent from Figure 8 are regions, particularly in the SH, where the bias reduction from clustering is negative; that is, the cluster-based MMM agrees less well with the observations than the simple arithmetic MMM. To understand this, one must consider that the clustering approach relies in some way on the density of model data points within the ensemble data

space. If data from a given model is less in agreement with the other models within the ensemble, but closer to the observed value, data from said model will be removed from the cluster-based MMM. While this is a limitation of the approach, it is also this feature of the clustering process that allows for the model spread of an ensemble to be readily investigated. For example, the clustering algorithm provides information regarding which models are included where and when in the MMM values (see below section). However, we note that the majority of the grid cells see a positive improvement in bias reduction through

clustering. For example, Figure 9 shows a binary map plot of areas where the bias reduction is positive (i.e improved, red), negative (worse, blue) and where there is no change (white). On an annual mean basis, >60% of grid-cells are 'improved', while ∼70% are improved or unchanged. Importantly, the magnitude of the positive bias reductions greatly exceed those of the negative changes as can be seen from the histogram given in Figure A2.

## 5.3 Insights from clustering into model agreement and spread

We investigate the degree to which individual models are included/excluded from the clustered MMM by counting the number of months where that model is used, at each location, as shown in Figure 10. This offers a simple mechanism to visualise model spread; outliers are more often excluded, models which fall in the pack are more often included. This information can be used together with Figure 7 as a means to identify areas warranting further investigation, and potentially priorities for





model development. For example, model **G** differs significantly from the cluster-based ensemble mean in the mid-latitude NH, over both land and ocean, as evidenced by the fact that it is virtually always excluded in this region. Similarly, Model **K** differs substantially from the other models in the SH, while model **N** is consistently different over South America in particular (potentially pointing towards a spurious model feature concerning ozone - e.g. regional precursor emissions). In the case

of model **K**, for example, it should be stressed that this does not necessarily suggest that the model is bad in the SH as a whole, merely that the model differs from the others. In fact, as was noted earlier, the cluster-based MMM agrees less well to observations in the SH, compared to the simple MMM, meaning that model **K** - which will have been excluded during the clustering process - could be closer to reality (observations) in this region, relative to the other models. This reasoning/approach potentially provides a useful framework to guide further investigation. We note that all models are included at some locations,

i.e. there is no blanket exclusion of certain models using these clustering techniques. In fact, some models e.g. models **C** and **J** are almost always included in the clustered MMM at each location, suggesting modelled ozone fields that are somewhat typical and in broad agreement with the ensemble mean.

## 6   Future work

While the principles presented here are robust and proven to be beneficial, some areas of methodological development/refinement

have been identified. For example, we currently assign all model data from the ensemble a cluster membership, used in turn to indicate which model data to include in the MMM. We have yet to consider the impact of weighting data by (a) distance from cluster centre and (b) distance from location of simple MMM (as opposed to a simple include/exclude rule). We also intend to explore the application of clustering in time, in addition to the mainly spatial methods presented here. Further, at present clusters are allowed to form in three dimensions, latitude, longitude and the predicted column ozone. In this way we

allow for a degree of uncertainty in the model output. Future work will build on this by developing methods to incorporate estimates of standard deviation and range associated with the modelled mean values into our techniques, thus enabling a more sophisticated treatment of uncertainty. Finally, forthcoming model intercomparison initiatives, e.g. CMIP6, will provide an excellent opportunity to apply our methods to consider parameters other than ozone that are of atmospheric interest (e.g. other short-lived climate forcing agents).

## 7   Concluding remarks

In this paper, we have investigated the applicability of advanced data clustering methods as an analytical/diagnostic tool with which to examine multi-model climate output. Relative to more rudimentary approaches, clustering offers a flexible method to calculate multi-model mean (MMM) fields of a given prognostic variable. The techniques operate by selectively includ-ing/excluding certain data at a given location, into the MMM calculation, based on the density of data points from the model

ensemble. The flexibility arises as the clustering methods examine the surrounding data space (e.g. spatially) to account for small spatial/mismatch errors (e.g. arising due to differing coarse model grids), thus offering an advantage over more tradi-



tional weighting methods. The techniques were applied to simulated fields of tropospheric column ozone from the 14 CCMs that took part in the ACCMIP model intercomparison. We demonstrate that cluster-based MMM tropospheric column ozone fields exhibit a lower bias with respect to observations, compared to a simple arithmetic MMM approach. On a global mean basis a reduction in the absolute bias is achieved through clustering in all months. In some months the bias reduction offered by

clustering is up to ∼16%. Additionally, we show that clustering offers a useful framework in which to readily identify and visualise model spread and outliers, on a regional basis. We suggest that such techniques could prove valuable in the identification of model development areas and provide insight surrounding regional strengths/deficiencies of specific models, or an ensemble as a whole, and to help characterise uncertainty. Finally, while we have focused on tropospheric ozone, we note that there is broad scope to develop the application of these techniques within the atmospheric sciences to examine other compounds of

climate-relevance.

*Code and data availability.* The clustering code, including demo software (Hyde, 2017) and related data sets, used to generate the results in this paper are available via GitHub: https://rhyde67.github.io/CATaCoMB-Climate-Model-Ensemble/. The latest release is available via Zenodo, DOI: 10.5281/zenodo.1119038. The model data files are available at the Centre for Environmental Data Analysis (CEDA): http://www.ceda.ac.uk/

*Competing interests.* The authors declare no competing interests.

*Acknowledgements.* This work was supported by the EPSRC through a pilot study (Advanced Data Clustering for Climate Science Applications, RFFLP027) as part of the Research on Changes of Variability and Environmental Risk (ReCoVER) program. R. Hossaini is also supported by a NERC Independent Research Fellowship (NE/N014375/1). We thank Paul Young for data access and helpful discussions.



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





**Table 1.** Observed and multi-model mean (MMM) global tropospheric ozone column (DU) between 60°N to 60°S latitude. Observations are a satellite-based climatology (Ziemke et al., 2011). Model data is from the historical (year 2000) ACCMIP simulation. The simple MMM is the arithmetic mean of all models, while the sigma mean excludes data outside of 1 standard deviation. For the cluster-based MMMs, results are shown for the 2 different permutations as outlined in Sect. 4.4.

|  | Jan | Feb | Mar | Apr | May | Jun | Jul | Aug | Sep | Oct | Nov | Dec | Annual Mean |
|---|---|---|---|---|---|---|---|---|---|---|---|---|---|
| Observation | 28.7 | 28.8 | 29.7 | 30.7 | 31.5 | 32.6 | 33.1 | 32.8 | 32.8 | 32.1 | 31.1 | 29.8 | 31.1 |
| Simple MMM | 29.4 | 29.5 | 30.1 | 30.4 | 30.7 | 31.4 | 31.9 | 32.2 | 32.3 | 31.4 | 30.1 | 29.5 | 30.7 |
| Sigma MMM | 29.0 | 29.2 | 29.9 | 30.2 | 30.4 | 31.1 | 31.7 | 32.0 | 32.0 | 31.3 | 30.1 | 29.4 | 30.5 |
| DDC (simple) MMM | 29.0 | 29.2 | 29.8 | 30.3 | 30.6 | 31.2 | 31.8 | 32.1 | 32.1 | 31.3 | 29.9 | 29.3 | 30.5 |
| DDC (sigma) MMM | 29.0 | 29.2 | 29.8 | 30.3 | 30.6 | 31.2 | 31.7 | 32.1 | 32.0 | 31.2 | 29.9 | 29.2 | 30.5 |





**Table 2.** Global monthly mean bias (DU) in tropospheric ozone column, see Eq. (1) between the various MMMs and observations presented in Table 1.

|  | Jan | Feb | Mar | Apr | May | Jun | Jul | Aug | Sep | Oct | Nov | Dec | Annual Mean |
|---|---|---|---|---|---|---|---|---|---|---|---|---|---|
| Simple MMM | 0.6 | 0.7 | 0.4 | -0.3 | -0.8 | -1.3 | -1.2 | -0.6 | -0.6 | -0.7 | -1.0 | -0.4 | -0.4 |
| Sigma MMM | 0.3 | 0.5 | 0.2 | -0.5 | -1.1 | -1.5 | -1.4 | -0.8 | -0.8 | -0.8 | -1.0 | -0.5 | -0.6 |
| DDC (simple) MMM | 0.3 | 0.4 | 0.1 | -0.4 | -0.9 | -1.4 | -1.3 | -0.7 | -0.8 | -0.9 | -1.2 | -0.6 | -0.6 |
| DDC (sigma) MMM | 0.2 | 0.4 | 0.1 | -0.4 | -0.9 | -1.4 | -1.4 | -0.7 | -0.8 | -0.9 | -1.2 | -0.6 | -0.6 |





**Table 3.** As Table 2 but the absolute bias (DU) according to Eq. (2).

|  | Jan | Feb | Mar | Apr | May | Jun | Jul | Aug | Sep | Oct | Nov | Dec | Annual Mean |
|---|---|---|---|---|---|---|---|---|---|---|---|---|---|
| Simple MMM | 3.5 | 3.9 | 3.8 | 3.7 | 3.7 | 3.4 | 3.1 | 3.0 | 3.9 | 4.4 | 4.2 | 3.8 | 3.7 |
| Sigma MMM | 3.2 | 3.6 | 3.7 | 3.7 | 3.7 | 3.5 | 3.2 | 3.1 | 3.9 | 4.4 | 4.1 | 3.7 | 3.6 |
| DDC (simple) MMM | 3.1 | 3.4 | 3.2 | 3.2 | 3.2 | 3.0 | 3.0 | 2.7 | 3.5 | 4.1 | 3.8 | 3.3 | 3.3 |
| DDC (sigma) MMM | 3.1 | 3.3 | 3.2 | 3.2 | 3.2 | 3.0 | 3.0 | 2.8 | 3.5 | 4.1 | 3.7 | 3.4 | 3.3 |





**Table A1.** Summary and citations for the ACCIP models/data sets used in this work.

| No. | Model Name | Citation |
|---|---|---|
| 1 | CMAM | Canadian Centre for Climate Modelling and Analysis (2011) |
| 2 | CICERO | Centre for International Climate and Environment Research - Oslo (2011) |
| 3 | EMAC | DLR German Institute for Atmospheric Physics (2011) |
| 4 | GFDL-AM3 | Geophysical Fluid Dynamics Laboratory (2011) |
| 5 | GISS-E2-R | NASA Goddard Institute for Space Studies (2011) |
| 6 | GEOSCCM | NASA Goddard Space Flight Center (2011) |
| 7 | CESM-CAM-superfast | Lawrence Livermore National Laboratory (2011) |
| 8 | LMDzORINCA | Laboratoire des Sciences du Climat et de l'Environnement (2011) |
| 9 | MOCAGE | Météo-France (2011) |
| 10 | NCAR-CAM-3.5 | NCAR (National Centre for Atmospheric Research) et al. (2011) |
| 11 | MIROC-CHEM | NCAS British Atmospheric Data Centre (2011) |
| 12 | UM-CAM | NIWA (2011) |
| 13 | STOC-HadAM3 | of Edinburgh (2011) |
| 14 | HadGEM2 | Hadley Centre for Climate Prediction and Research (2011) |



**Figure 1.** Principles of the cluster-based ensemble mean method illustrated using a synthetic data set. (a) A synthetic spatially-varying observation (X). (b) Predictions of X from 4 idealised models (see main text). (c) Cluster analysis of the model data sets using the DDC clustering algorithm. Ellipses represent the different clusters that are formed, and the black crosses are outliers not included in the clusters. (e) Comparison of the multi-model mean (MMM) of X derived from either a simple arithmetic mean of all model data (red) or one based on clusters (green). Observation data from panel (a) is again shown in black.





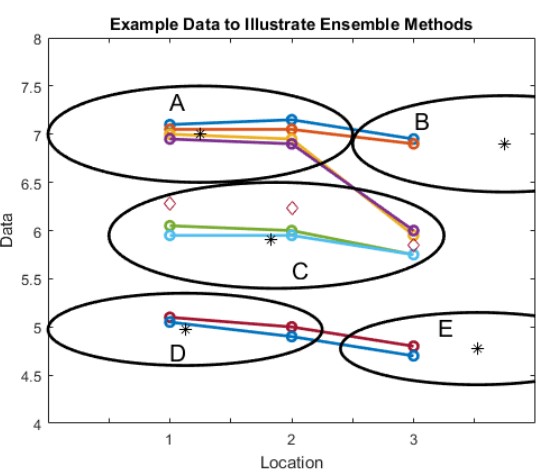

**Figure 2.** Synthetic data used to illustrate the different ensemble methods. Model data is represented by the coloured lines with markers. The red diamonds are the predicted truth and the asterisks are cluster centres.





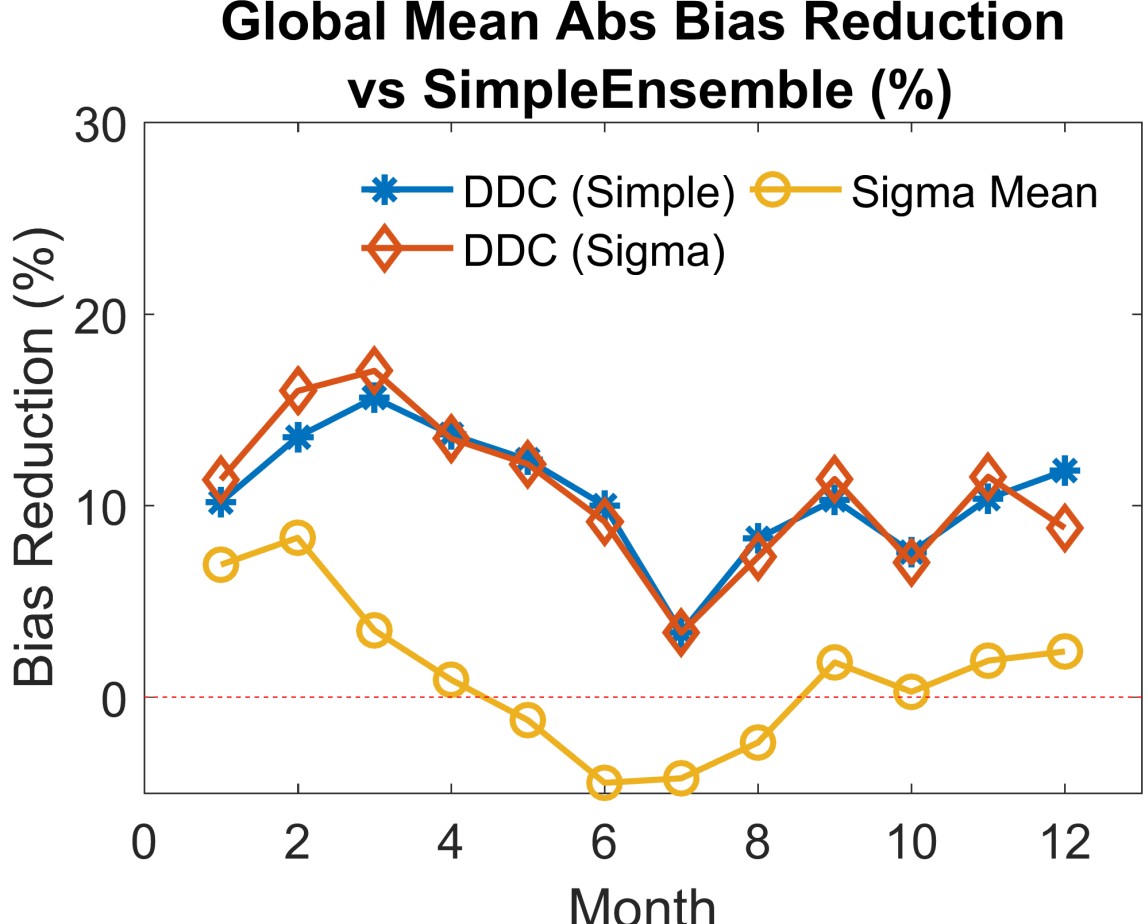

**Figure 3.** Monthly variability in global mean absolute bias reduction (%, MMM ozone - observed ozone) with respect to simple arithmetic MMM. Blue points denote improvement gained using DDC (predicted truth = Simple) to determine model inclusion into the MMM. Red points denote improvement from DDC (predicted truth = Sigma) . Orange points denote improvement gained using just the model spread (1 Sigma) to determine model inclusion into the MMM (i.e. without clustering).





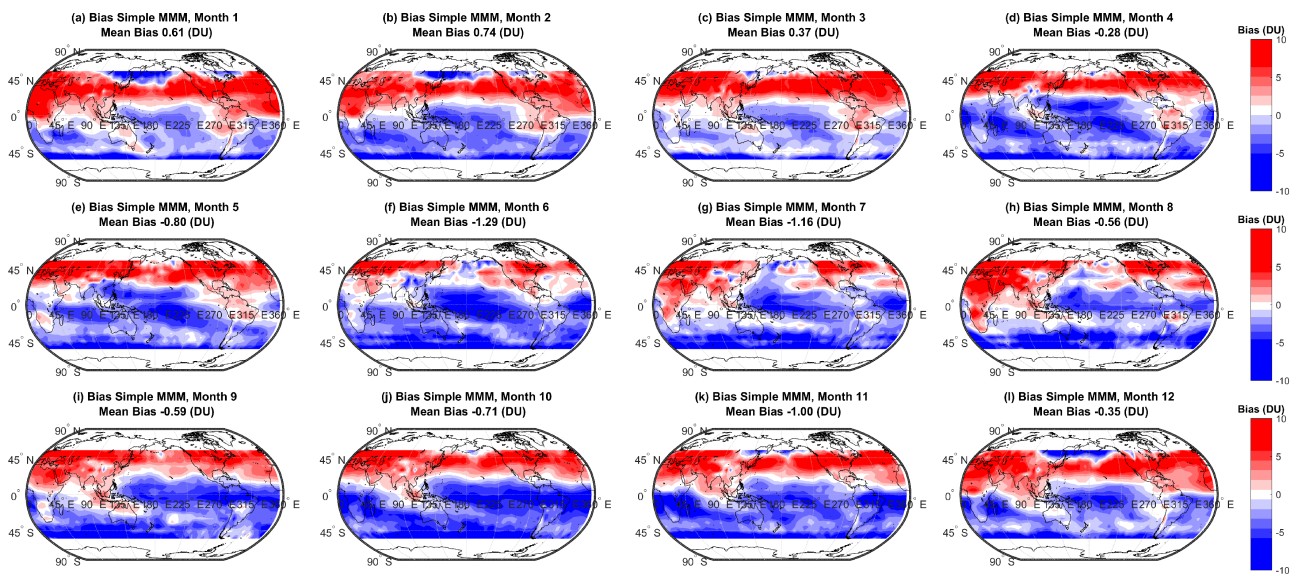

**Figure 4.** Monthly bias (DU) between the simple arithmetic multi-model mean (MMM) tropospheric ozone column and the observed climatology.

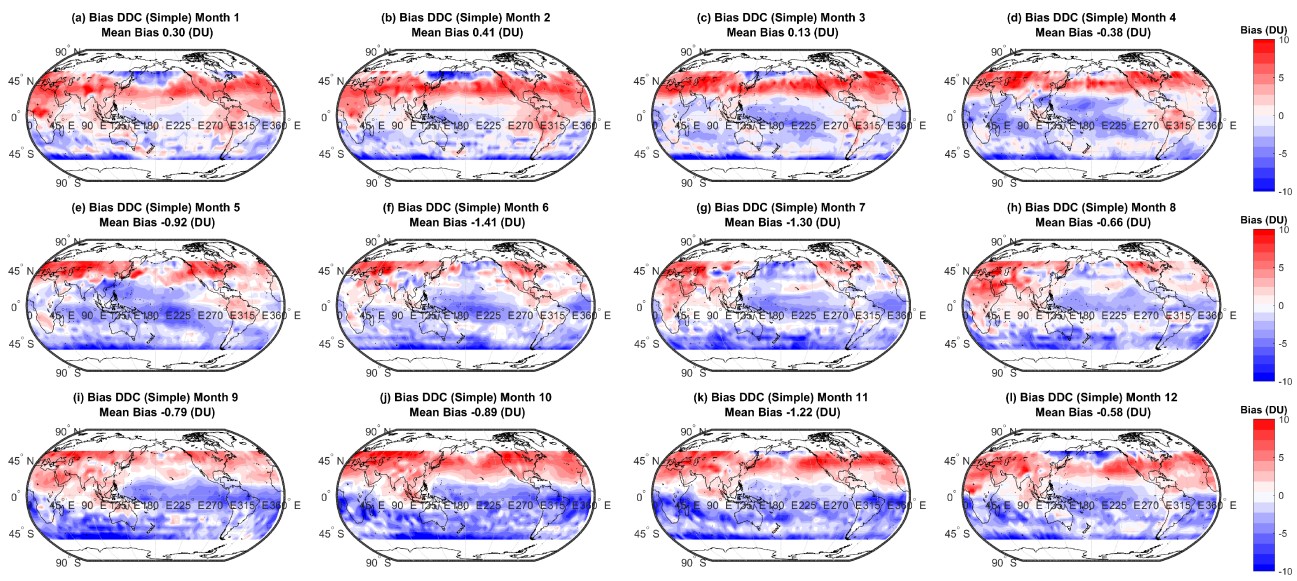

**Figure 5.** As Figure 4 but for the cluster-based MMM (simple).





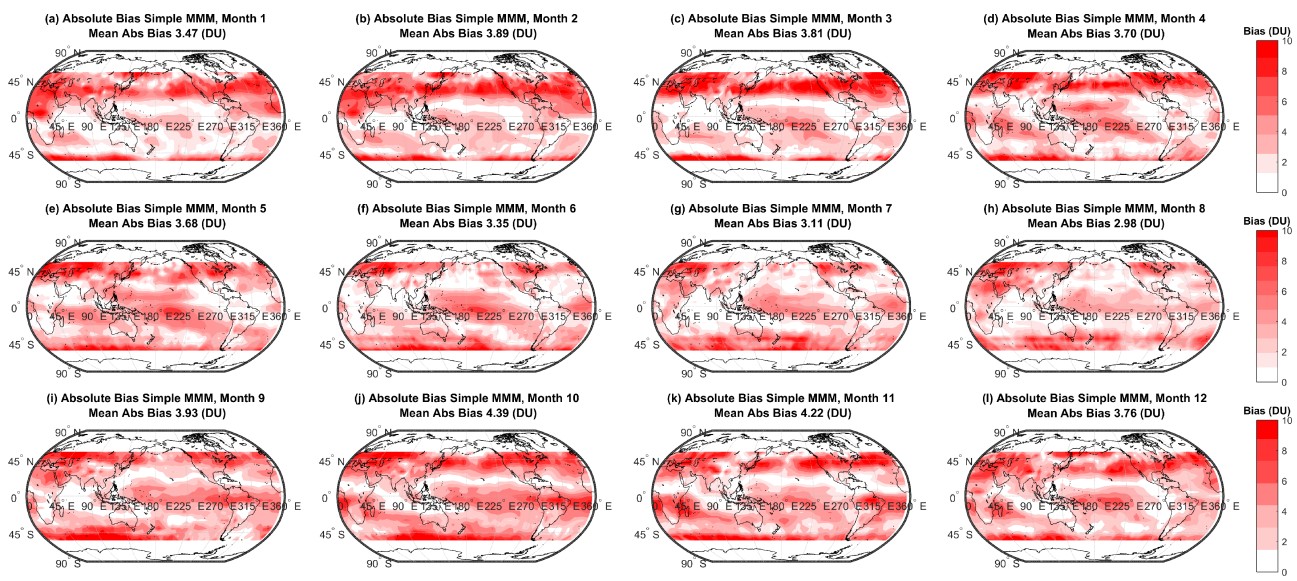

**Figure 6.** Monthly absolute bias (DU) between the simple arithmetic multi-model mean (MMM) tropospheric ozone column and the observed climatology.

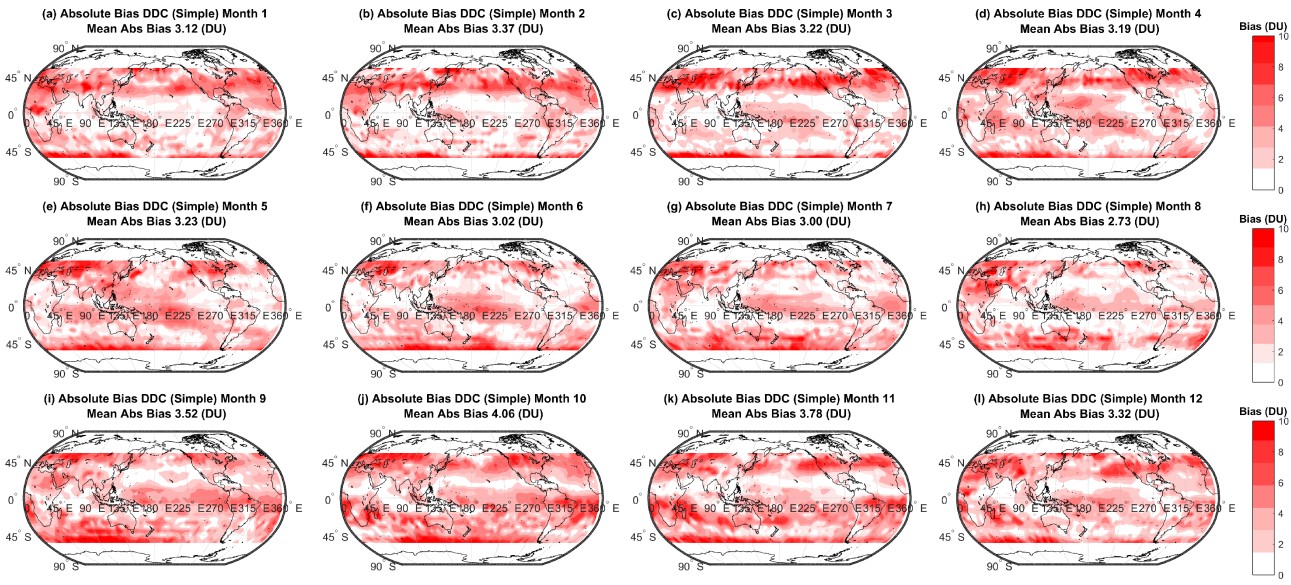

**Figure 7.** As Figure 6 but for the cluster-based MMM (simple).



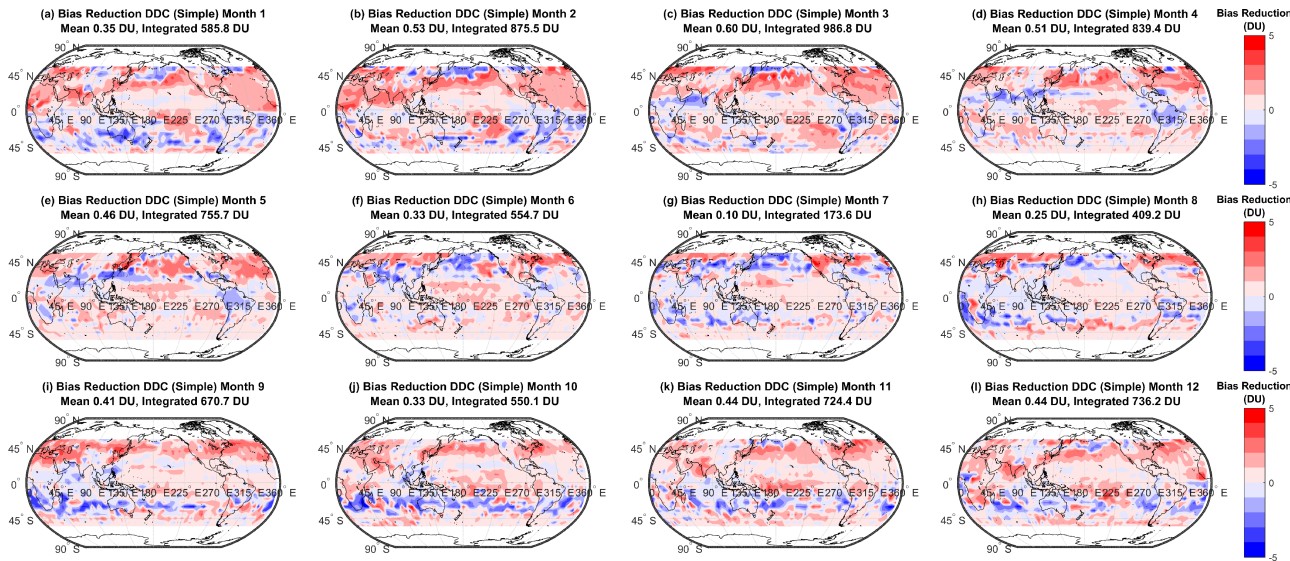

**Figure 8.** Monthly bias reduction (DU) defined as the difference in the absolute bias between the cluster-based MMM ozone column and observations, and the simple arithmetic MMM and observations. Where the bias reduction is positive (i.e. red) indicates areas where the cluster-based MMM agree better with the observations than the simple arithmetic MMM. In the title of each panel, the global mean absolute bias reduction, and the absolute bias reduction summed over all grid-cells are shown.

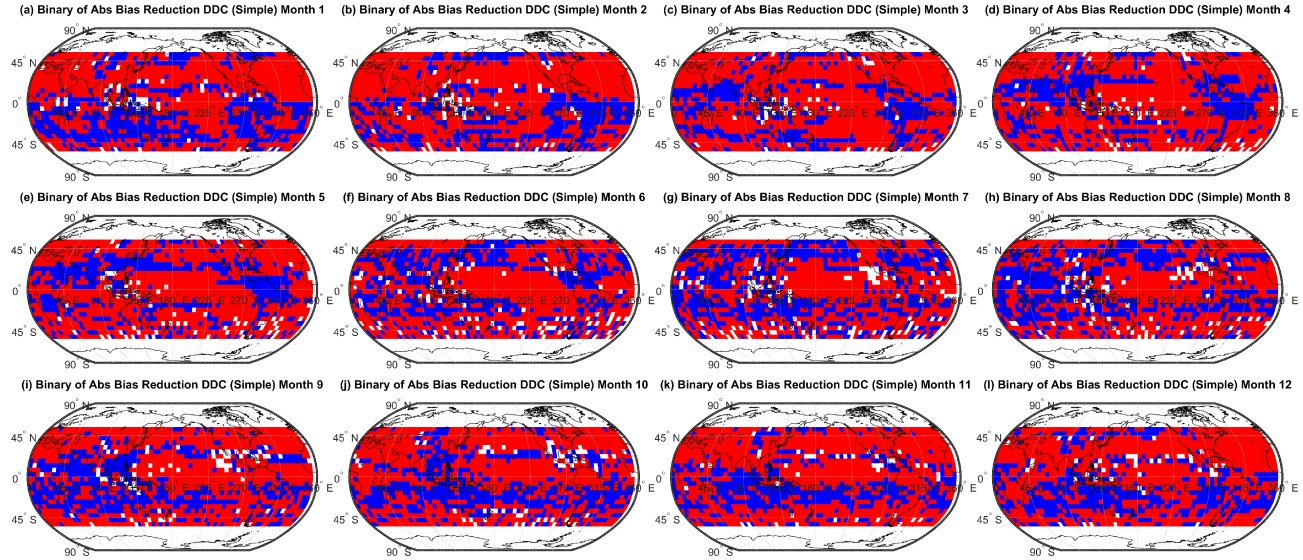

**Figure 9.** As Figure 8 but showing a binary of grid-cells in which the MMM has improved (red) or got worse (blue) as a result of clustering (unaffected cells filled white).





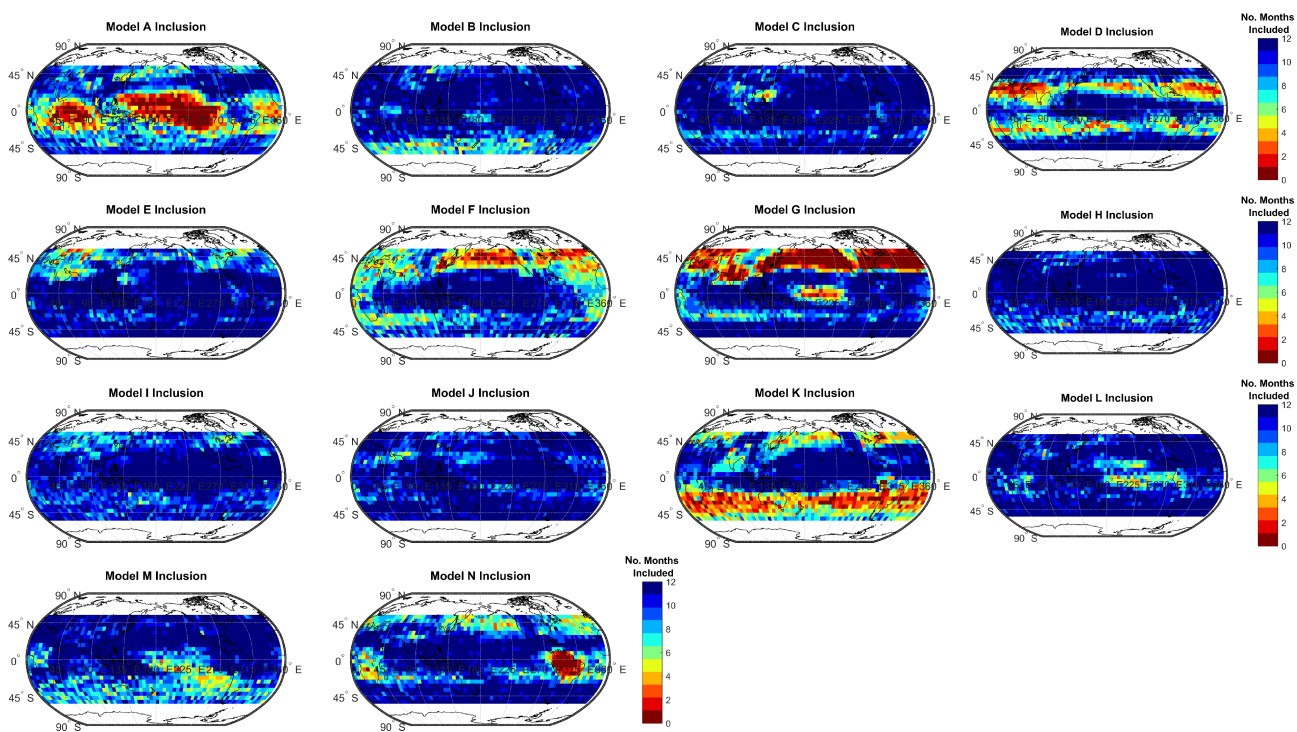

**Figure 10.** Number of months each model (names removed, labelled A-N) are included in the cluster-based MMM. For a given region, models that are seldom included (i.e. a low numbers of months) differ from the ensemble pack.





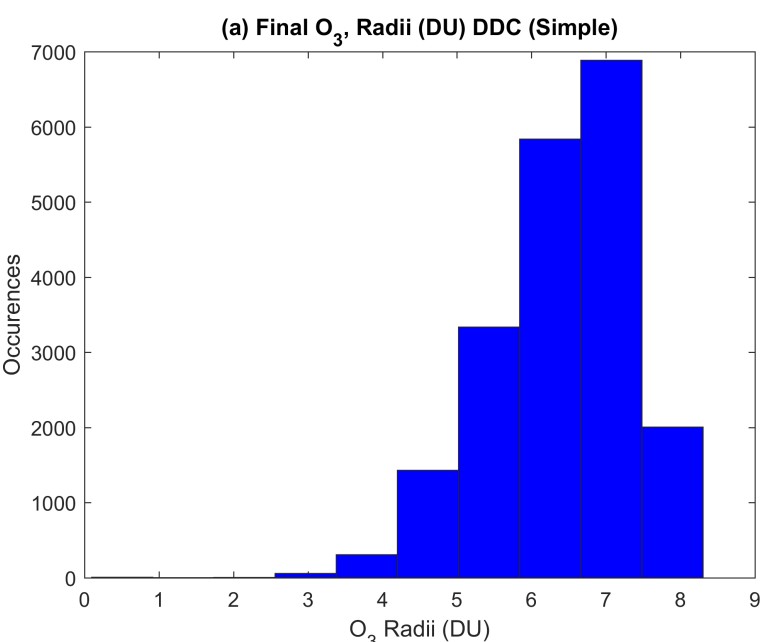

**Figure A1.** Radii in the ozone dimension (DU) for clusters used to calculate the MMM.



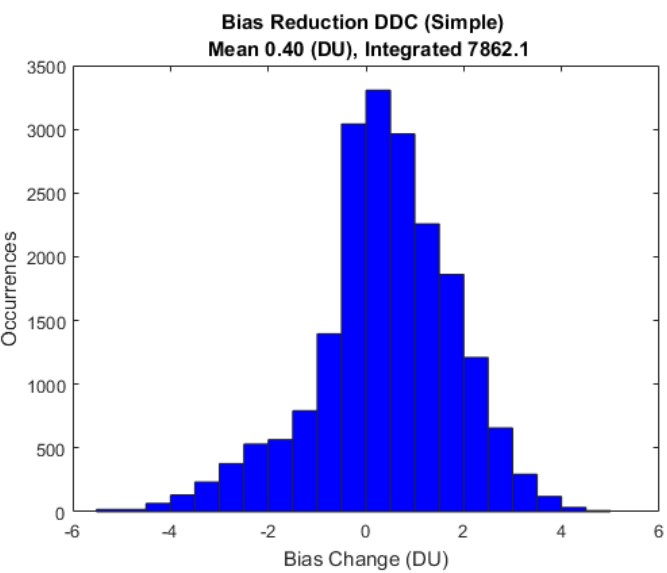

**Figure A2.** Magnitude of the yearly column ozone bias reduction due to clustering; clustered MMM vs observations relative to simple MMM vs observations, see Sect. 5.2.