# Peer review of "Cluster-based analysis of multi-model climate ensembles"

_Geoscientific Model Development, 2017_

## Referee Comment (RC1) · Anonymous Referee #1 · 12 Feb 2018

Summary: As an alternative to simpler multi-model means, the authors explore the advantages to applying a clustering technique to the analysis of ACCMIP modelled tropospheric ozone columns. They generally find that the multi-model mean agrees better (i.e. causes a reduction in the mean absolute bias) if generated using a clustering technique than when either a straight arithmetic averaging or even a weighting based on standard deviation are used. The improvement occurs not only in the global average but also in a majority of grid cells, particularly in polluted areas. Clustering is not often used in atmospheric science, but is more common in other areas. The authors suggest that there is scope for greater uptake of this analysis technique in areas such as atmospheric science.

The paper offers an interesting, not entire new but underused method to produce multimodel means. En route, it also produces by-products, such as highlighting regions where a model is or is not used to produce the mean. This information can be used to direct model development.

The paper is well written; the language and graphics are appropriate for ACP. Upon studying the paper, I have found no significant problems that require correction. Below are only two semantic issues. I recommend publication of the paper subject to addressing these minor issues.

Details:

P3l20: The word "hyper-elliptical" requires explanation. In the atmospheric sciences, few people will know what that means.

P8l12: Replace "greater to" with "greater than".

---

## Referee Comment (RC2) · M.G. Schultz (Referee) · 13 Feb 2018

The manuscript by Hyde et al. presents a novel application of an efficient clustering method to the analysis of multi-model ensemble results. As the authors rightly point out, there have been relatively few applications of clustering to atmospheric data yet, and the clustering method allows for some deeper insights into large datasets, which do not reveal their secrets if only simple traditional methods, i.e. multi-model means are applied. The paper is well written and the details of the method and results are well presented. I also strongly support the idea to explore such methods in the context of climate data with an objective to improve the analyses of multi-model results and model evaluation in general. However, I must raise two fundamental concerns which I have with this study and which prevent me from recommending this article for publication.

These concerns are explained below. Because of the fundamental nature of these concerns I refrain from making detailed technical comments. I strongly encourage the authors to revise their manuscript and put the study on a solid foundation. As a community we clearly need this kind of method development! But we also have to make sure that we are not violating good practices when it comes to scrutinizing model results in confrontation with observations.

Concern #1: From the outset, the aims of the study are formulated as trying to reduce the bias between a multi-model ensemble and observations – in this case a climatology of tropospheric column ozone retrieved from a satellite instrument. For example, the abstract states "As a proof of concept, we show the proposed clustering technique can offer improvement in terms of reducing the absolute bias between the MMMs and observations." The problem I have with this is that the study objective therefore mixes two very different things: a clustering method, which has the sole purpose of classifying data into groups of somehow similar properties, and a model-data comparison, which should try to objectively describe how well or how poorly the models agree with observations and put this in relation to estimates how well the models could agree with the observations in theory, i.e. answering the question what degree of certainty a model-data comparison can give us. By mixing the objectives of these two steps and using the reduced bias as a proof of concept for applying cluster analysis to ensemble model results, the study loses the objectivity that is required for a meaningful model evaluation.

Concern #2: I never heard of a requirement to know the "predicted truth" in a cluster analysis, and this concept actually frightens me, because it suggests that the analysis is performed in such a way that the desired results are obtained by tweaking the method until it fits. First, there is no principle need to enforce selection of one cluster after the grouping is done – this actually defeats the purpose of clustering! Second, if the authors do wish to pick one (or two?) clusters from the resulting groups, then they should apply accepted, robust statistical methods for this. In the introduction a suggestion was

made to select the cluster with most members. This would be an objective criterion. In terms of model evaluation it would convey the message that one selected a certain estimate of the truth from the majority of models. This must be clearly separated from the value that is represented by that cluster. In fact, it could be very valuable to define measures that describe how many clusters are formed from the model ensemble in each region or throughout time. For example, one could use the ratio of the number of members in the most populated cluster over the number of members in the next populated cluster – if this value is large, then this means that most models agree with each other; if the ratio approaches 1, the models differ a lot among themselves, and this indicates that some process is not well understood or well represented in all models. This conclusion can be reached regardless of any comparison with observations.

To illustrate these issues, let me present a simple example for a single point in the model evaluation (e.g. a single grid column value of a single month). Assume that the retrieval yields a TCO value of 30 DU. Five models produce values of 50 DU, and five models produce values of 10 DU. The traditional MMM will generate an average of 30 DU from these and claim a perfect match between model and observation. But the MMM hides the fact that none of the models was even close to the measurement, and it is by pure chance that the average of all models agrees with the data (had we had only nine models, we would have found a bias). Without additional analysis, we thus believe that our models are perfect. The cluster technique represents one way of taking the analysis a step further: in this example, it will identify two groups of model results, one at 10 DU and one at 50 DU. This is it. That is all that the cluster analysis does, and this is the information that should be used to describe the quality of the model results (see example of the ratio measure above). What the authors now suggest is to use a "predicted truth" to identify one of the clusters. In this hypothetical example, this would either fail (because both clusters are equally far away from the "truth"), or one of the clusters would be randomly selected. In the latter case, one might actually obtain an "evaluation" result which looks reasonable on the surface, because the large bias of 20 DU would be identified. However, with a slight modification of the example, i.e.

by adding a few models, say 2 with a result of 30 DU to the ensemble, the proposed method would identify these models as a third cluster and suggest perfect agreement with the observations. Clearly, this result is not meaningful in the context of evaluating the "quality" of the models or of the model ensemble.

Another issue which is not reflected in the manuscript is the error of the observations. A recent study by Gaudel eta al. (Elementa – Science of the Anthropocene, under review) shows substantial deviations between different TCO retrievals. Again, in order to make a robust statement about the quality of a model, one needs to know what a deviation between the model result and an observed value actually means. Would a perfect model even be expected to exactly reproduce the observation? (see also Solazzo et al., 2016/2017 on AQMEII evaluation)

Finally, I would like to emphasize again that I very much welcome the introduction of clustering as an analysis technique for model results, and that I see great potential for this method. As the authors describe, the technique is robust (or should I say graceful) against typical errors in the data sampling step of the model evaluation, i.e. if a feature is shifted by one grid box. The grouping of data within the model ensemble indeed offers several new insights to the "behavior" of models. My sole criticism is: what the authors claim is a "proof of concept" actually disproves the concept, because it forces the concept of clustering to achieve something it is not meant to do.

---

## Author Comment (AC1) · 6 Apr 2018

**"Cluster-based ensemble means for climate model intercomparison" by Hyde et al.**

**Authors' response to Reviewer 2 (Dr Martin Schultz)**

We thank Dr Schultz for his detailed and insightful comments on our work. We are extremely pleased that he finds the clustering approach to be a "novel" and "forward-looking" method, supports the ideas behind our paper, and finds the manuscript well-written and well-presented.

We acknowledge some of the more substantive concerns raised by Dr Schultz and have worked to address these comments, noting the strong encouragement from Dr Schultz to revise the manuscript given its potential to the field. We are confident that these comments are easily addressed in a revision to the manuscript and that the manuscript is strengthened accordingly.

In particular, we have reframed the paper to reflect the fact that we are exploring the application of clustering to climate model ensembles in a broad sense, rather than specifically attempting to derive an *improved* estimate of the ensemble multi-model-mean (MMM).

Our responses are given in blue, with Dr Schultz's comments repeated in bold.

**Concern #1: From the outset, the aims of the study are formulated as trying to reduce the bias between a multi-model ensemble and observations – in this case a climatology of tropospheric column ozone retrieved from a satellite instrument.**

Our intention in performing this study was to 'investigate the applicability of advanced data clustering methods as an analytical/diagnostic tool with which to examine multi-model climate output' (as stated in the Conclusions). We appreciate that this does not fully come across clearly in the early part of the paper and so we will reframe the Abstract and Introduction of the paper in a revised manuscript to better reflect this.

**For example, the abstract states "As a proof of concept, we show the proposed clustering technique can offer improvement in terms of reducing the absolute bias between the MMMs and observations." The problem I have with this is that the study objective therefore mixes two very different things: a clustering method, which has the sole purpose of classifying data into groups of somehow similar properties, and a model-data comparison, which should try to objectively describe how well or how poorly the models agree with observations and put this in relation to estimates how well the models could agree with the observations in theory, i.e. answering the question what degree of certainty a model-data comparison can give us.**

**By mixing the objectives of these two steps and using the reduced bias as a proof of concept for applying cluster analysis to ensemble model results, the study loses the objectivity that is required for a meaningful model evaluation.**

The objective of the study was as indicated in our response to the first part of this comment above. As part of this we explored the extent to which clustering could be used to *refine* a multi-model ensemble by excluding clear outliers, we also explored the *membership* of the resulting clusters in terms of space and time. We assessed the impact of this ensemble refinement on model-observation agreement and the degree to which this could be used to identify model development priorities to be explored in future work; for example, why a specific model may always be excluded from the dominant cluster at a particular location.

It was not our intention to 'use the reduced bias as a proof of concept for applying cluster analysis', rather we intended to show that reducing model-observation bias was found to be a consequence of our ensemble refinement. Therefore, in the revised manuscript we have

reframed the text significantly and reworded sentences such as the reviewer highlights above, to better reflect our intentions.

**Concern #2: I never heard of a requirement to know the "predicted truth" in a cluster analysis, and this concept actually frightens me, because it suggests that the analysis is performed in such a way that the desired results are obtained by tweaking the method until it fits.**

On reflection we feel that 'predicted truth' was perhaps not the best choice of phrase. Our intention was that the value in question is more like an 'initial guess' which we then refined and we certainly did not use this value to drive our results. We will rephrase this throughout in the revised manuscript.

**First, there is no principle need to enforce selection of one cluster after the grouping is done – this actually defeats the purpose of clustering!**

Since one of our objectives was to investigate the impact of using clustering to exclude outliers from an ensemble multi-model-mean (MMM), we feel that we did need to adopt a method of cluster selection in order to define the membership of our ensemble sub-sample.

**Second, if the authors do wish to pick one (or two?) clusters from the resulting groups, then they should apply accepted, robust statistical methods for this. In the introduction a suggestion was made to select the cluster with most members. This would be an objective criterion.**

The arithmetic multi-model-mean (i.e. the average of all models) is a commonly used metric for analysing climate model output and something that the community is familiar with. As such, we chose to use this value in selecting our primary cluster. We acknowledge that cluster selection based on population (i.e. choosing the most populous) would be more objective and so have adapted our approach accordingly in the revised manuscript.

Since ensemble members tend to cluster around the all-model MMM, we find that the most populous cluster is also the cluster that contains the all-model MMM in 98% of cases. As such, adapting our sub-sample selection strategy has had very little impact on our results. In terms of the reduction in model-obs bias between the all-model MMM and sub-sample MMM, we find that when using the most populous cluster to define the sub-sample, the model-obs bias is reduced at 64.6% of places/times, unchanged at 3.9% of places/times and slightly increased at 31.4% of places/times. This is in good agreement with our earlier results (initial submission), using the cluster containing the all-model MMM, which found bias reduction in 60% of places/times, no change at 10% and a slight increase in bias at 30% of place/times.

Spatially, we see no significant change to model-obs bias reduction/increase (see Figure 1 below) and we also see no significant change to model inclusion/exclusion (see Figure 2 below) which means that our comments on these results in the manuscript still hold.

Our revised manuscript will reflect these changes, in both the Method and Results section, which will we hope will be satisfactory to the editor and Dr Schultz.

[Figure]

***Figure 1:*** *Reproduction of Figure 8 in the initial manuscript, except using the most populous cluster to generate the ensemble sub-sample as opposed to the cluster containing the all-model MMM.*

[Figure]

***Figure 2:*** *Reproduction of Figure 10 in the initial manuscript, except using the most populous cluster to generate the ensemble sub-sample as opposed to the cluster containing the all-model MMM.*

**In terms of model evaluation it would convey the message that one selected a certain estimate of the truth from the majority of models. This must be clearly separated from the value that is represented by that cluster.**

Again, we feel that this is a semantic issue. We did not intend to imply that the 'predicted truth' was the truth, more that it was an 'initial guess' at what the truth might be. This will be clarified in the revised manuscript.

**In fact, it could be very valuable to define measures that describe how many clusters are formed from the model ensemble in each region or throughout time. For example, one could use the ratio of the number of members in the most populated cluster over the number of members in the next populated cluster – if this value is large, then this means that most models agree with each other; if the ratio approaches 1, the models**

differ a lot among themselves, and this indicates that some process is not well understood or well represented in all models.

This conclusion can be reached regardless of any comparison with observations. To illustrate these issues, let me present a simple example for a single point in the model evaluation (e.g. a single grid column value of a single month). Assume that the retrieval yields a TCO value of 30 DU. Five models produce values of 50 DU, and five models produce values of 10 DU. The traditional MMM will generate an average of 30 DU from these and claim a perfect match between model and observation. But the MMM hides the fact that none of the models was even close to the measurement, and it is by pure chance that the average of all models agrees with the data (had we had only nine models, we would have found a bias). Without additional analysis, we thus believe that our models are perfect. The cluster technique represents one way of taking the analysis a step further: in this example, it will identify two groups of model results, one at 10 DU and one at 50 DU. This is it. That is all that the cluster analysis does, and this is the information that should be used to describe the quality of the model results (see example of the ratio measure above). What the authors now suggest is to use a "predicted truth" to identify one of the clusters. In this hypothetical example, this would either fail (because both clusters are equally far away from the "truth"), or one of the clusters would be randomly selected. In the latter case, one might actually obtain an "evaluation" result which looks reasonable on the surface, because the large bias of 20 DU would be identified. However, with a slight modification of the example, i.e. by adding a few models, say 2 with a result of 30 DU to the ensemble, the proposed method would identify these models as a third cluster and suggest perfect agreement with the observations. Clearly, this result is not meaningful in the context of evaluating the "quality" of the models or of the model ensemble.

We thank Dr Schultz for his in depth explanation of this point and we agree with his suggestion to include an analysis of the ratio between number of members in the most vs second-most populous cluster.

We are currently working on this analysis for inclusion in the revised manuscript but present initial findings here.

Figure 3 shows a histogram of the ratio between the number of members in the second most populous cluster (cluster 2 hereafter) and the number of members in the most populous cluster (cluster 1 hereafter) at all points in space/time. A small number indicates that there is a significant difference, i.e. that the most populous cluster has >> members than the second most populous. This suggests that the model spread is sufficiently small for most models to be included in cluster 1, and thus the models that are excluded from this cluster can be considered outliers. Conversely, if this number is large, this suggests that model spread is larger in these locations/times. As such, both cluster 1 and cluster 2 can probably be considered equivocal in terms of representing the ensemble.

As can be seen from Figure 3, in the majority of cases we consider, cluster 1 has significantly more members that cluster 2. This confirms that, in the majority of cases, sub-sampling the ensemble based on the membership of cluster 1 can be considered to be robust. It is important to note however that there is tail of data points with ratio values => 0.5 for which sub-sampling based on cluster 1 is less reasonable.

[Figure]

**Figure 3:** *Histogram of ratio (x-axis) of number of members in second most populous cluster (cluster 2) to most populous cluster (cluster 1)*

We assess the degree to which the ratio between number of members in cluster 2 and cluster 1 varies in space and time (Figure 4). Our initial assessment suggests that the higher ratio values tend to occur in the mid-latitudes (suggesting greater model spread), with tropical locations exhibiting lower cluster ratios in general. There also appears to be some seasonality to the signal; higher ratios (thus greater model spread) are more likely to occur during the summer months. It is interesting to note that regions where the ratio > 0.5 seems, by eye, to coincide with regions where the model-observation bias is increased when the ensemble is sub-sampled to the membership of cluster 1. This suggests that by excluding data here we are in fact removing data points which are in closer agreement with the observations.

[Figure]

**Figure 4:** *Spatial and temporal variability in ratio of number of members in second most populous cluster (cluster 2) to most populous cluster (cluster 1).*

In future work we will look at the possibility of using clustering to generate a weighted MMM, where ensemble members are weighted according to their cluster membership, i.e. members of the most populous cluster contributing more to the MMM than the less populous clusters and clear outliers.

We will elaborate on these points and include some further discussion in the revised manuscript. We hope that this will be satisfactory to the editor and Dr Schultz.

**Another issue which is not reflected in the manuscript is the error of the observations. A recent study by Gaudel eta al. (Elementa – Science of the Anthropocene, under review) shows substantial deviations between different TCO retrievals. Again, in order to make a robust statement about the quality of a model, one needs to know what a deviation between the model result and an observed value actually means. Would a perfect model even be expected to exactly reproduce the observation? (see also Solazzo et al., 2016/2017 on AQMEII evaluation)**

We agree with Dr Schultz that the error in the observations is important. Indeed we identify this in the manuscript as something that we will address in further work. We feel that in this manuscript we are not explicitly making statements about model quality, rather we are showcasing the potential applications of the clustering method to this type of data. As this work moves forward we will be certain to develop a treatment of both observation and model uncertainty, however for now we feel that it is beyond the scope of this particular manuscript.

---

## Author Comment (AC2) · 6 Apr 2018

**"Cluster-based ensemble means for climate model intercomparison" by Hyde et al.**

Authors' response to Reviewer 1

We thank the reviewer for his/her comments on our manuscript and are pleased that he/she supports the work, finding it "interesting" and "well-written".

We will address both of the semantic issues highlighted by the reviewer in the revised manuscript.

---

## Author Response (AR1)

**Dr Ryan Hossaini**
NERC Independent Research Fellow

Lancaster Environment Centre,
University of Lancaster,
Lancaster, LA1 4YW, UK.

**Email:** r.hossaini@lancaster.ac.uk

**RE: Revisions to Manuscript (gmd-2017-317)**

Dear Dr Fyke,

     Thank you for considering our paper, '**Cluster-based analysis of multi-model climate ensembles**', for publication in *GMD*. As noted in the response to reviewers, we were extremely pleased to receive such helpful comments from Dr Schultz (referee #2) in his thorough review. While Dr Schultz raised some concerns (addressed and discussed below), we were delighted that (a) he described our work as "novel" and "forward-looking" and (b) that Dr Schultz "strongly encouraged" us to revise the manuscript.

Dr Schultz correctly raised two "concerns" regarding the original submission. We believe the 2nd of these concerns was the more substantive, though both have been addressed in the revised submission. Below we wish to stress some key points regarding our revisions.

- **Concern #1** related to the aims of the study, and whether our paper was confusing presentation of an original (clustering) method with a model-data comparison. This largely presentational issue was easily addressed, and we have reframed the Abstract and Introduction to the manuscript to address the concern. Specifically, the aim of our study was to investigate the applicability of a novel (data clustering) method as an analytical/diagnostic tool with which to examine multi-model climate output. We are confident that this comes across more clearly in the revised manuscript. For further clarity and to reinforce the point, we have also amended the manuscript title to that given above.

Our paper applies cluster analysis to output of tropospheric ozone from multiple climate models. By then selecting certain clusters, the model ensemble can be subsampled, for example, to generate a 'cluster-based' multi-model mean ozone field – devoid of outliers identified in the clustering process.

- **Concern #2** related to how best this subsampling should be performed to ensure a fully automated and objective selection. In the revised manuscript, we have adopted Dr Schultz's recommendation of choosing the most populous cluster. This slight change in method had a minimal impact on our findings. We also performed the additional analysis suggested by Dr Schultz by examining the ratio of the number of members in the most populous cluster to the number in the 2nd most populous cluster. This analysis is discussed in a new section (Sect. 5.4) accompanied by two new figures (Figs. 9 and 10).

In summary, we believe that we have addressed Dr Schultz's comments fully and robustly. In doing so, we also believe that the manuscript has been strengthened. Once again thank you for considering our manuscript.

Please let us know if any further action is required and we look forward to hearing from you.

Yours sincerely,

*R Hossaini*

**Ryan Hossaini**

[Figure]

**Dr Ryan Hossaini**
NERC Independent Research Fellow

Lancaster Environment Centre,
University of Lancaster,
Lancaster, LA1 4YW, UK.

**Email:** r.hossaini@lancaster.ac.uk

**RE: Revisions to Manuscript (gmd-2017-317)**

Dear Dr Fyke,

Thank you for considering our paper, '**Cluster-based analysis of multi-model climate ensembles**', for publication in *GMD*. As noted in the response to reviewers, we were extremely pleased to receive such helpful comments from Dr Schultz (referee #2) in his thorough review. While Dr Schultz raised some concerns (addressed and discussed below), we were delighted that (a) he described our work as "novel" and "forward-looking" and (b) that Dr Schultz "strongly encouraged" us to revise the manuscript.

Dr Schultz correctly raised two "concerns" regarding the original submission. We believe the 2nd of these concerns was the more substantive, though both have been addressed in the revised submission. Below we wish to stress some key points regarding our revisions.

- **Concern #1** related to the aims of the study, and whether our paper was confusing presentation of an original (clustering) method with a model-data comparison. This largely presentational issue was easily addressed, and we have reframed the Abstract and Introduction to the manuscript to address the concern. Specifically, the aim of our study was to investigate the applicability of a novel (data clustering) method as an analytical/diagnostic tool with which to examine multi-model climate output. We are confident that this comes across more clearly in the revised manuscript. For further clarity and to reinforce the point, we have also amended the manuscript title to that given above.

Our paper applies cluster analysis to output of tropospheric ozone from multiple climate models. By then selecting certain clusters, the model ensemble can be subsampled, for example, to generate a 'cluster-based' multi-model mean ozone field – devoid of outliers identified in the clustering process.

- **Concern #2** related to how best this subsampling should be performed to ensure a fully automated and objective selection. In the revised manuscript, we have adopted Dr Schultz's recommendation of choosing the most populous cluster. This slight change in method had a minimal impact on our findings. We also performed the additional analysis suggested by Dr Schultz by examining the ratio of the number of members in the most populous cluster to the number in the 2nd most populous cluster. This analysis is discussed in a new section (Sect. 5.4) accompanied by two new figures (Figs. 9 and 10).

In summary, we believe that we have addressed Dr Schultz's comments fully and robustly. In doing so, we also believe that the manuscript has been strengthened. Once again thank you for considering our manuscript.

Please let us know if any further action is required and we look forward to hearing from you.

Yours sincerely,

*R Hossaini*

**Ryan Hossaini**

[revised manuscript text omitted]